# Magnetoelectric Effect in Amorphous Ferromagnetic FeCoSiB/Langatate Monolithic Heterostructure for Magnetic Field Sensing

**DOI:** 10.3390/s23094523

**Published:** 2023-05-06

**Authors:** L. Y. Fetisov, M. V. Dzhaparidze, D. V. Savelev, D. A. Burdin, A. V. Turutin, V. V. Kuts, F. O. Milovich, A. A. Temirov, Y. N. Parkhomenko, Y. K. Fetisov

**Affiliations:** 1Research and Educational Center ‘Magnetoelectric Materials and Devices’, MIREA—Russian Technological University, 119454 Moscow, Russia; m.v.dzhaparidze@mail.ru (M.V.D.); dimsav94@gmail.com (D.V.S.); phantastic@mail.ru (D.A.B.); fetisov@mirea.ru (Y.K.F.); 2Laboratory of Physics of Oxide Ferroelectrics, National University of Science and Technology MISiS, 119049 Moscow, Russia; aturutin92@gmail.com (A.V.T.); viktor.kuts.3228@yandex.ru (V.V.K.); filippmilovich@mail.ru (F.O.M.); temirov.alex@yandex.ru (A.A.T.); parkh@rambler.ru (Y.N.P.)

**Keywords:** magnetoelectric effect, monolithic heterostructure, magnetic field sensor, metglas, magnetron sputtering, langatate, magnetostrictive films

## Abstract

This paper investigates the possibilities of creating magnetic field sensors using the direct magnetoelectric (ME) effect in a monolithic heterostructure of amorphous ferromagnetic material/langatate. Layers of 1.5 μm-thick FeCoSiB amorphous ferromagnetic material were deposited on the surface of the langatate single crystal using magnetron sputtering. At the resonance frequency of the structure, 107 kHz, the ME coefficient of linear conversion of 76.6 V/(Oe∙cm) was obtained. Furthermore, the nonlinear ME effect of voltage harmonic generation was observed with an increasing excitation magnetic field. The efficiency of generating the second and third harmonics was about 6.3 V/(Oe^2^∙cm) and 1.8 V/(Oe^3^∙cm), respectively. A hysteresis dependence of ME voltage on a permanent magnetic field was observed due to the presence of α-Fe iron crystalline phases in the magnetic layer. At the resonance frequency, the monolithic heterostructure had a sensitivity to the AC magnetic field of 4.6 V/Oe, a minimum detectable magnetic field of ~70 pT, and a low level of magnetic noise of 0.36 pT/Hz^1/2^, which allows it to be used in ME magnetic field sensors.

## 1. Introduction

Currently, magnetic field sensors of various types are widely used in different fields of science and technology, including navigation, electronics, geophysics, healthcare, and others [1]. Detecting DC and AC magnetic fields involves using effects of electromagnetic induction, magnetization of ferromagnetics, the Hall effect in semiconductors, magnetoresistance and magnetoimpedance phenomena, and various optical and quantum effects.

Over the past two decades, a bulk of research has extensively focused on the development of new types of sensors that use the magnetoelectric (ME) effect in composite heterostructures containing ferromagnetic (FM) and piezoelectric (PE) layers [2,3]. The ME effect manifests itself in the generation of an electric voltage by the structure under the action of a magnetic field resulting from the deformation of the FM layer and, as a consequence, polarization of the mechanically linked PE layer [4]. ME sensors offer several advantages over magnetic sensors of other types, including high sensitivity, simple construction, low power consumption, and the use of film manufacturing technologies. Additionally, ME magnetic field sensors can compete with SQUID magnetometers, which require cooling to the temperature of liquid helium, whereas ME sensors have high sensitivity to AC and DC magnetic fields at room temperature [5].

Studies have shown that to increase the sensitivity to the magnetic field of ME sensors, FM layers of heterostructures should be made from materials with a high value of the piezomagnetic coefficient *q* (amorphous metallic alloys), and PE layers should be made from materials with a high ratio of piezoelectric modulus to dielectric permittivity *d*/*ε* (piezoceramics, single crystals, piezopolymers) [5]. It is important for the layers of a structure to have high acoustic quality, which can increase the sensitivity of sensors by two to three orders of magnitude when detecting magnetic fields whose frequency coincides with the acoustic resonance frequency of the structure [6,7]. Since single crystals have the highest acoustic quality, studying the ME effects in structures with single crystal layers is of significant interest. By now, ME effects have been studied in heterostructures with PE layers made of quartz [8], langatate [9], lead magnoniobate-titanate [10], gallium arsenide [11,12], and lithium niobate [13,14,15], produced by layer bonding. The presence of a glue layer in the structure led to a decrease in acoustic quality and, accordingly, the magnitude of the resonant ME effect. In recent work [16], the behavior of organic and inorganic glues for high-temperature operations of an ME sample based on a 0.364BiScO_3_–0.636PbTiO_3_ piezoelectric ceramic and Terfenol-D alloy was investigated. It was shown that the organic glue-based ME composite sharply decreases the output signal with increasing temperature, while the inorganic glue-based ME composite gradually decreases the ME effect (by 20% less at 200 °C than at room temperature). The use of sputtering technique for the ME composite structure eliminates losses related to glue or other connecting layers. The highest efficiency of ME conversion was achieved in monolithic heterostructures with PE layers made of aluminum nitride [17], produced by magnetron sputtering. 

This study aims to investigate the possibility of creating ME magnetic field sensors based on a monolithic heterostructure containing a langatate (LGT) monocrystal plate, on whose surface a thin film of amorphous ferromagnet (AmF) is deposited. The langatate monocrystal was chosen due to its relatively high ratio of piezomodulus to dielectric permittivity *d/ε* ≈ 0.25, high acoustic quality factor *Q* ~ 5 × 10^4^, and no pyroeffect. Another fact is that it undergoes no phase transitions up to the melting temperature of 1450 °C [18]. Amorphous alloy FeCoSiB [13,19] was selected as the material for the FM layer because it has high magnetostriction, saturates in small magnetic fields, and has a small hysteresis.

The first part of the paper describes the studied heterostructure and measurement methods. Then, the measured characteristics of linear and nonlinear ME effects in the heterostructure are presented; the obtained results and characteristics of the magnetic field sensor using the linear ME effect are discussed. The main research results are summarized in the Conclusion.

## 2. Materials and Methods

The measurements were performed on a symmetric three-layer composite structure AmF-LGT-AmF, schematically shown in Figure 1.

A single crystal of La_3_Ga_5.5_Ta_0.5_B_10_ with an X-cut was grown using the Czochralski method (Fomos Materials, Moscow, Russia) and had a plate shape with dimensions of 20 mm × 4 mm in the plane and a thickness of *a_p_* = 0.6 mm. The piezoelectric module of the crystal was *d*_31_ ≈ 5.2 pm/V, and the relative dielectric constant was *ε* ≈ 22 [9,18]. Layers of amorphous ferromagnetic alloy Fe_70_Co_8_Si_12_B_10_, with a thickness of 1.5 μm each, were deposited on both surfaces of the LGT plate. Conductive AmF layers were used as electrodes. The structure was mounted in the measurement setup on thin wires, which were connected to the electrodes using silver paste.

The amorphous ferromagnet layers were deposited by the high-frequency (13.56 MHz) magnetron sputtering method using the SUNPLA-40TM equipment (Seoul, Republic of Korea) at room temperature and a power of 200 W. The Fe_70_Co_8_Si_12_B_10_ target with a diameter of 50 mm was utilized for deposition. The target was sputtered in an ionized argon atmosphere at a pressure of 0.5 Pa. The film deposition rate was ~1 nm/min.

The structural characterization of the AmF alloy films was performed using the JEM 2100 (JEOL, Tokyo, Japan) transmission electron microscope (TEM) at an accelerating voltage of 200 kV. The samples were thinned using the focused ion beam (FIB) technique with the Strata FIB 201 System (FEI Company, Hillsboro, OR, USA). Figure 2 shows a bright-field TEM image of the edge of the sample lamella. The crystalline nanosized particles are clearly visible in the image, and their micro-X-ray spectral analysis revealed a reliable match with the α-Fe phase. In addition to the crystalline α-Fe particles, an amorphous phase was present in the film. This was inferred from the presence of an amorphous halo on the electronogram along with the α-Fe reflections.

The block diagram of the measurement setup for studying ME effects in a heterostructure is shown schematically in Figure 1. Measurements were carried out using the method of harmonic modulation of the magnetic field at room temperature. The structure was placed in a permanent magnetic field *H* = 0–120 Oe, directed along the long axis of the sample, which was created using the Helmholtz coils, with a diameter of 12 cm, powered by a TDK Lambda GENH600-1.3 DC power supply. A variable magnetic field *h*cos(2π*ft*) with an amplitude of up to *h* ≈ 0.06 Oe and a frequency of *f* = 0–150 kHz was applied parallel to the permanent field, which was created using a second pair of 5 cm diameter Helmholtz coils connected to an Agilent 33,210 generator. The variable voltage *u*(*f*) generated by the structure was measured using a lock-in SR844 (Stanford Research Systems, Sunnyvale, CA, USA) with an input impedance of 1 MΩ. The noise spectrum of the structure was measured using an Agilent E4448A spectrum analyzer (Keysight Technologies, Santa Rosa, CA, USA). The magnitude of the permanent field *H* was measured using a LakeShore 421 gaussmeter (Lake Shore Cryotronics, Inc., Westerville, OH, USA). The amplitude of the variable field *h* was controlled by the current through the modulating coils, which were calibrated at a frequency of 100 Hz. Dependencies of the voltage amplitude *u* generated between the LGT surfaces were measured as a function of the permanent field *H*, frequency *f*, and amplitude *h* of the excitation field. Frequency spectra of the voltage were obtained using fast Fourier transform. The setup’s control was ensured by the dedicated LabVIEW12 software on a PC.

## 3. Results

### 3.1. Linear ME Effect

At the first stage, the linear characteristics of the direct ME effect in the described structure were measured at low excitation magnetic field amplitudes. Figure 3 shows the dependence of the ME voltage *u* on the frequency *f* of the excitation field with an amplitude of *h* = 0.06 Oe at a bias field of *H* = 25 Oe. Only one peak was present on the characteristic near the frequency *f*_0_ ≈ 106.95 kHz with an amplitude *u*_1_ = 250 mV and a quality factor *Q = f/*Δ*f* = 2380, where Δ*f* is the peak width at the level of 0.71. As shown below, the peak corresponds to the excitation of the lowest resonance mode of longitudinal acoustic vibrations in the structure. Resonance on bending vibrations in the structure was absent due to the symmetric arrangement of the magnetic layers. Based on a series of frequency response curves that helped determine the value of the resonance voltage *u*_1_, the dependencies below were constructed as functions of the DC and AC magnetic fields.

The measured dependence of the ME voltage *u*_1_ on the resonance frequency from the applied DC magnetic field *H* is shown in Figure 4. The *u*_1_(*H*) dependence features a classical form: the voltage initially increases with the growth of *H*, reaches a maximum of ~266 mV at the field *H*_m1_ ≈ 25 Oe, and then monotonically decreases as the ferromagnetic layers saturate. The shape of the curve is determined by the shape of the field dependence of the piezomagnetic module of the FM layer, where *λ*(*H*) is the dependence of the magnetostriction λ on the static field [20]. The dependence shows hysteresis when the direction of the applied magnetic field is reversed to the opposite with a coercive force of *H*_c1_ ≈ 3 Oe.

During the variation of the DC magnetic field *H*, a slight change in the resonant frequency of the structure *f*_0_ and a significant change in the *Q* factor were observed. The frequency *f*_0_ changed by ~0.04%, reaching a minimum value at the field *H*_m1_. The quality factor *Q* decreased from *Q* ≈ 4200 at *H* = 0 Oe to *Q* ≈ 2380 at the optimal field *H*_m1_ and then increased again to 4200 with the magnetic field increasing up to 100 Oe. Figure 5 shows the dependence of the ME voltage *u*_1_ at the resonant frequency *f*_0_ on the amplitude of the excitation magnetic field *h* at the optimal field *H*_m1_. The dependence is linear in the range of magnetic fields from ~0.7 μOe to ~0.03 Oe. The tangent of the slope of the dependence *u*_1_*/h*, which characterizes the sensitivity of the structure to the AC magnetic field, was ≈4.6 V/Oe. The inset in Figure 5 shows the frequency response curve measured at the minimum excitation amplitude of *h* = 50 pT. The noise level of the measuring circuit (*u* = 0.46 μV) is at least an order of magnitude lower than the signal level from ME sample at the resonant frequency. However, the voltage amplitude (2.8 μV) of the ME signal at a minimal excitation amplitude of 50 pT does not fit into the linear dependence of the response of the ME sensor.

### 3.2. Nonlinear ME Effect

The nonlinear ME effect of resonance voltage harmonic generation was studied at the increased amplitude of the excitation field *h*. The structure was excited by a harmonic field with a frequency equaling a multiple of the resonant frequency *f* = *f*_0_/*n* (where *n* = 2, 3...). The spectra of the generated voltage were recorded. Figure 6 shows the spectra of ME voltage at excitation frequencies of (a) *f* = *f*_0_/2 = 53.48 kHz (at *h*_2_ = 0.11 Oe, *H*_2_ = 3 Oe) and (b) *f* = *f*_0_/3 = 35.36 kHz (at *h*_3_ = 0.16 Oe, *H*_3_ = 7 Oe). In both cases, the frequency spectrum contains a component at the excitation frequency and a component at the resonant frequency *f*_0_. Thus, when the structure is excited by a field with a frequency of *f*_0_/2, the second order voltage harmonic with amplitude *u*_2_ is generated, and when excited by a field with a frequency of *f*_0_/3, the third order voltage harmonic with amplitude *u*_3_ is generated.

Figure 7 shows the dependencies of the amplitudes of the voltage harmonics *u*_2_ and *u*_3_ on the magnetic field *H* in the nonlinear mode. The shapes of the curves are determined by the shape of the *n*th derivative of magnetostriction with respect to the magnetic field $\lambda^{\left(n\right)}\left(H\right)={\partial^{n}\lambda /\partial H^{n}|}_{H}$ [21]. Both dependencies showed hysteresis in both the magnetic field and voltage amplitude. The coercive force for the second harmonic was *H*_c2_ ≈ 21 Oe; for the third harmonic, it was *H*_c3_ ≈ 4.5 Oe.

Figure 8 demonstrates the dependencies of the amplitudes *u*_2_ and *u*_3_ of the harmonics on the excitation field *h*. The value of the DC field *H* for each curve corresponds to the maximal voltage on the field dependencies (Figure 7): *H*_m2_ = 3 Oe and *H*_m3_ = 7 Oe, respectively. In the range of excitation fields from 0 to 0.1 Oe, the amplitude of the second harmonic grows quadratically *u*_2_ ~ *h*^2^, while the amplitude of the third harmonic grows as *u*_3_ ~ *h*^3^. The power-law dependence of the amplitudes of the harmonics on the exciting field *u_n_* ~ *h^n^* agrees with [22].

### 3.3. Sensor Characteristics

The sensitivity to AC magnetic field $S=u_{1}/h$ and the frequency dependence of noise level are important characteristics of the ME monolithic heterostructure described for its application in magnetic field sensors. Table 1 compares the sensitivities of structures with different compositions of single-crystal PE layers and AmF layers in the resonant mode.

As indicated by the table, the sensitivity of the investigated structure is comparable to or exceeds the sensitivities of other structures with amorphous magnetic alloys, except for the structure based on a PMN-PT single crystal [10]. Figure 5 shows that the minimal detectable field for this structure, determined from the condition of equality between the amplitude of the generated voltage and the noise level, is *h*_m_ ≈ 0.7 μOe or 70 pT. The equivalent magnetic noise density (EMND) is an important characteristic of the structure, which characterizes the magnetic field sensor based on it. The EMND is determined as the spectral density of the voltage noise generated by the structure divided by its sensitivity to AC magnetic fields. The voltage noise spectral density was measured using an Agilent E4448A spectrum analyzer with an average self-noise level of −152 dBm, a bandwidth of 2 kHz, and at room temperature without magnetic shielding of the structure in the absence of a permanent magnetic field. The sample was placed in an aluminum housing that provided electrostatic shielding. The resulting dependence of the spectral density of magnetic noise on the frequency for the investigated structure is shown in Figure 9. At the resonance frequency of *f*_0_ = 106.95 kHz, the level of magnetic noise of the structure sharply decreased and reached 0.36 pT/Hz^1/2^.

## 4. Discussion

We can calculate the acoustic resonance frequency of the structure using the formula for the fundamental mode of planar oscillations of a free rod [28]: $f_{r}=1/\left(2L\right)\sqrt{Y/\rho }$, where *L* is the length of the sample, and Y and ρ are the effective Young’s modulus and density of the structure, respectively. Since the thickness of the AmF layers is much smaller than the thickness of the LGT layer, their influence on the resonance frequency can be neglected. Substituting the known values *Y*_11_ = 110 GPa and =6130 kg/m^3^ for langatate into the formula, we obtain a resonance frequency of *f*_r_ ≈ 105.9 kHz, which matches the measured value of *f*_0_ ≈ 106.95 kHz (see Figure 3).

A shift in the acoustic resonance frequency of the structure *f*_0_ was observed when the field *H* was changed, with δ*f/f*_0_ ≈ 0.04%, due to the change in the Young’s modulus of the magnetostrictive layer in the magnetic field [20]. The small magnitude of the shift is explained by the presence of the AmF layers and non-ideality of fixing of the ME sample.

At the same time, the ferromagnetic layer, despite its small thickness, significantly affected the acoustic quality factor of the monolithic structure. As noted earlier, the *Q*-factor of the structure decreased by a factor of 1.76, from 4200 at *H* = 0 Oe to 2380 at the optimal bias field *H*_m1_. The *Q* of the structure at the field *H*_m1_ is comparable to quality factor *Q* ≈ 2200–2600 of similar LGT-Metglas structures containing a 25 μm-thick AmF layer and fabricated by bonding [22,29].

The maximum value of the ME coefficient for the direct linear resonance effect in the AmF-LGT-AmF structure, calculated from the data in Figure 5, was $\alpha_{E}^{\left(1\right)}=\frac{u_{1}}{a_{p}\cdot h}$= 76.6 V/(Oe·cm). The obtained ME coefficient was about 6 times smaller than the coefficient of 450 V/(Oe·cm) for the LGT-Metglas structure with magnetic layers of 20 μm thickness [29] and about 9 times smaller than the coefficient of 720 V/(Oe·cm) for the LGT-FeCoV structure with a magnetic layer thickness of 160 μm [9]. This is primarily due to the small thickness of the FM layer in the presented ME monolithic heterostructure, which was approximately eight times smaller than that in the LGT-Metglas structure fabricated by bonding [29]. However, the sensitivity 4.4 V/Oe for presented ME monolithic heterostructure is much higher than 1 V/Oe in similar LGT-Metglas described in ref. [25].

The values of the nonlinear ME coefficients, which determine the efficiency of second and third voltage harmonic generation, for the described ME monolithic heterostructure were $\alpha_{E}^{\left(2\right)}=u_{2}/\left(a_{p}\cdot h^{2}\right)$ = 6.3 V/(Oe^2^∙cm) and $\alpha_{E}^{\left(3\right)}=u_{3}/\left(a_{p}\cdot h^{3}\right)$ = 1.8 V/(Oe^3^∙cm), respectively, as shown in Figure 7 and Figure 8. Thus, the magnitude of the nonlinear coefficient for the monolithic heterostructure was only two times smaller than the nonlinear coefficient of 12 V/(Oe^2^∙cm) for the LGT-Metglas structure with a magnetic layer thickness of 25 μm, produced by the bonding of layers [30].

The specific feature of the ME effect in the described structure is the large value of the field *H*_m1_ ≈ 25 Oe, corresponding to the maximum ME coefficient, and the presence of hysteresis in the dependencies of the main and higher harmonics’ voltage on the bias field (Figure 4 and Figure 7) with a coercive field *H*_c_ ≈ 3–20 Oe. In structures with thin ferromagnetic films, where demagnetization effects are not a factor, the value of the field *H*_m_ is normally 2–5 Oe, and the hysteresis field for single-phase films does not exceed a fraction of an Oe [13,31]. The appearance of hysteresis is obviously related to the hysteresis of the dependence of magnetostriction on the magnetic field *λ*(*H*). Structural studies of the produced AmF films demonstrated the presence of the α-Fe crystalline phase. Such a phase can create residual magnetization in the material, which will magnetize the soft magnetic phase and give a non-zero ME effect without an external magnetic field [32]. It has been previously shown [24] that the α-Fe phase can be created in an amorphous alloy using the laser rapid annealing method. In this work, we have obtained the crystalline α-Fe phase directly in the process of depositing magnetic films.

The obtained value of EMND 0.36 pT/Hz^1/2^ for the proposed ME structure is comparable to the noise level of 1 pT/Hz^1/2^ in the AlN/Metglas structure at the resonance frequency of 860 Hz [25] and lower than the noise level of 12 pT/Hz^1/2^ in the LiNbO_3_/Metglas structure at a resonance frequency of 3 kHz [27] and 60 pT/Hz^1/2^ in the AlScN/Metglas structure at a resonance frequency of 8 kHz [24]. However, the EMND is highly dependent on frequency, and in future works, the frequency modulation technique can be used to increase sensitivity to low frequency magnetic field [33].

In conclusion, it should be noted that the high thermo-stability of the acoustic and dielectric properties of LGT enables sensors based on the monolithic AmF-LGT-AmF heterostructure to exhibit insignificant (less than 0.1%) temperature-dependent shifts in the resonance frequency and makes them suitable for use in the temperature range of −70 °C to +70 °C [29].

## 5. Conclusions

Thus, this study investigated the direct ME effect in a three-layered monolithic heterostructure consisting of an amorphous FeCoSiB/langatate laminate. The amorphous ferromagnetic layers were deposited onto the surface of a langatate crystal using magnetron sputtering. At the planar acoustic resonance frequency of the structure, the linear ME conversion coefficient was 76.6 V/(Oe·cm). Upon increasing the excitation magnetic field, a nonlinear ME effect of voltage harmonics generation was observed. The field dependence of the ME voltage exhibited hysteresis, which was attributed to the presence of the α-Fe crystalline phase in the magnetic film. The ME monolithic heterostructure showed a sensitivity of 4.6 V/Oe to the AC magnetic field at the acoustic resonance frequency and a minimum detectable field of ~70 pT, allowing it to be used for magnetic field sensors. The advantages of resonant field sensors based on langatate crystal structures are their high temperature stability and low magnetic noise density of ~0.36 pT/Hz^1/2^.

## Figures and Tables

**Figure 1 sensors-23-04523-f001:**
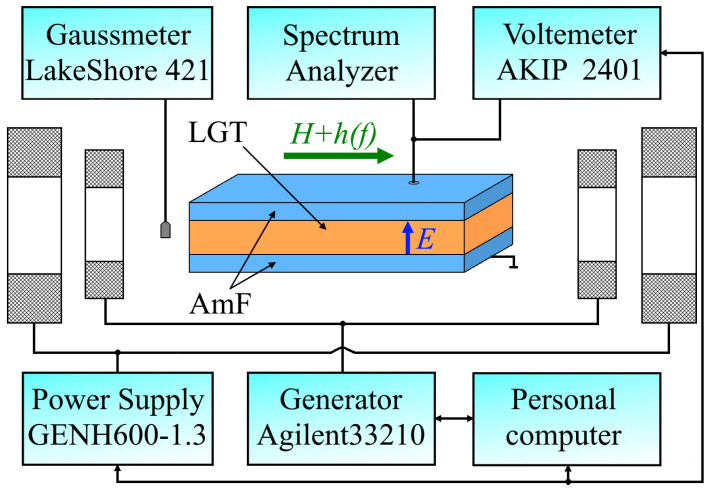
Block diagram of the setup for studying the ME effect.

**Figure 2 sensors-23-04523-f002:**
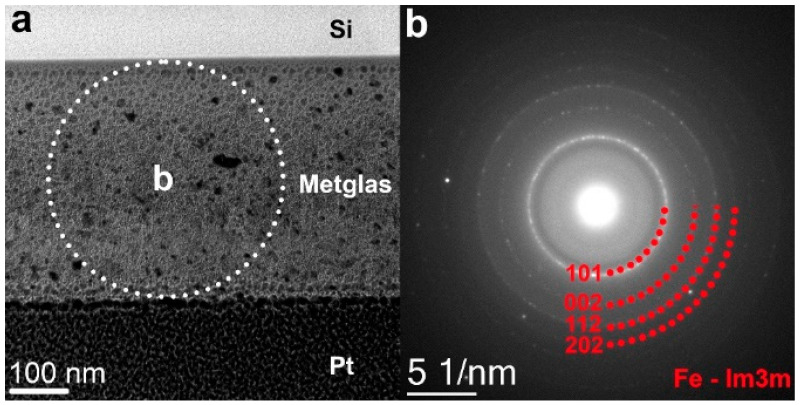
(**a**) TEM image and (**b**) electron diffraction pattern of the FeCoSiB film showing the locations of X-ray microanalysis (dotted red line). The bright-field image indicates the area from which the electron diffraction pattern was formed.

**Figure 3 sensors-23-04523-f003:**
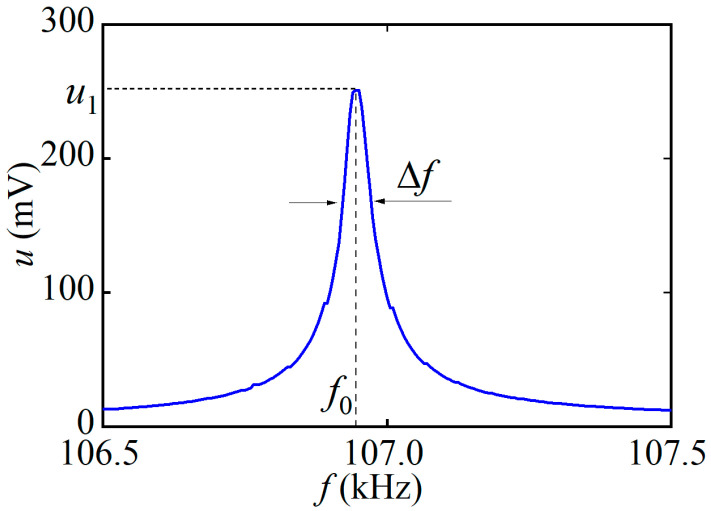
Dependence of the ME voltage *u* on the frequency *f* of the excitation magnetic field with the amplitude *h* = 0.06 Oe at *H* = 25 Oe.

**Figure 4 sensors-23-04523-f004:**
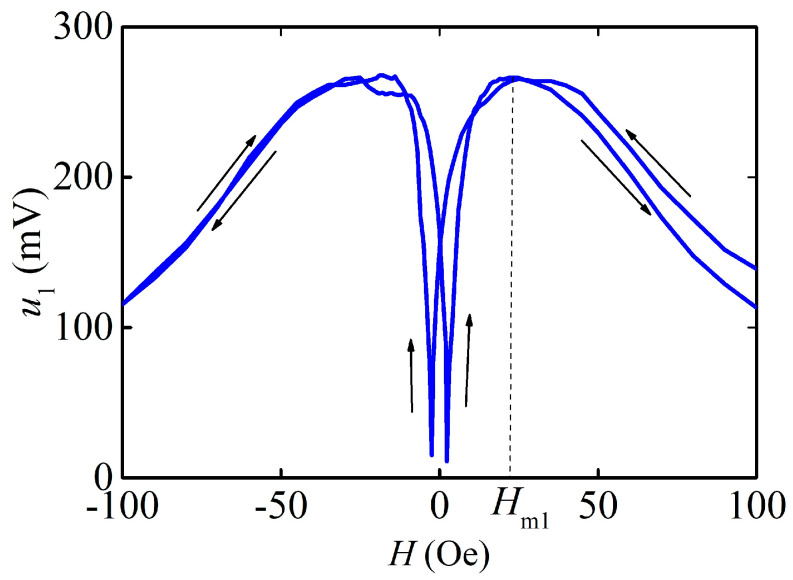
Dependence of the ME voltage *u*_1_ at the resonance frequency on the field *H* at *h* = 0.06 Oe. The arrows show the direction of the field variation. Arrows denote the direction of field change.

**Figure 5 sensors-23-04523-f005:**
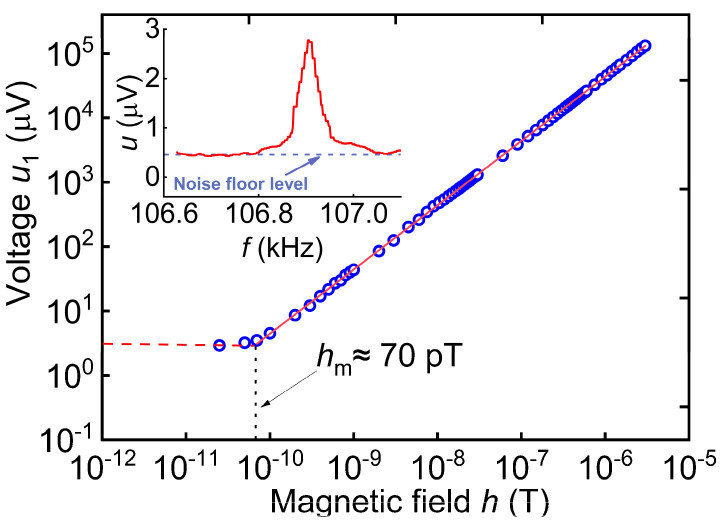
Dependence of the ME voltage *u*_1_ at the resonance frequency on the excitation field *h* at *H* = 25 Oe. The dashed red line is a linear approximation, and blue dots are experimental data.

**Figure 6 sensors-23-04523-f006:**
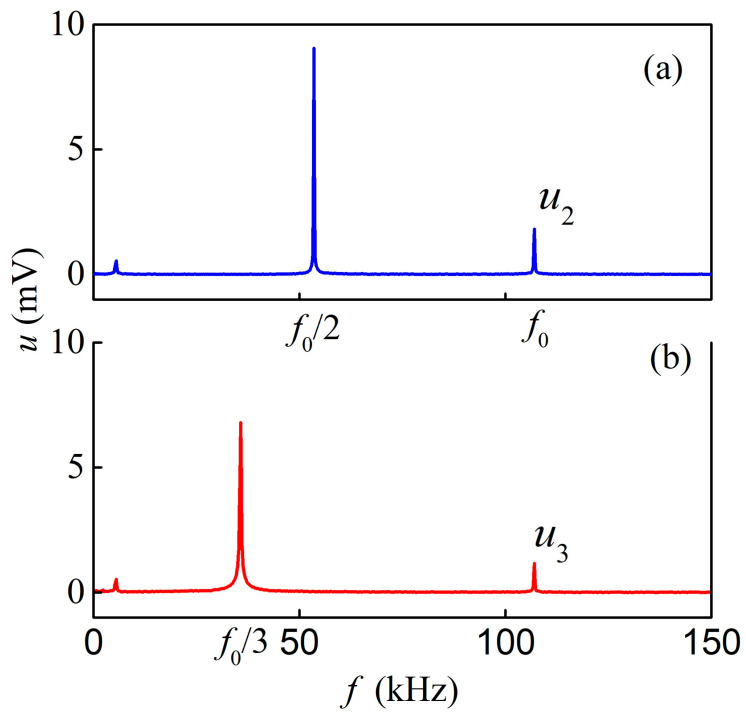
Frequency spectra of the ME voltage upon excitation of the AmF-LGT-AmF structure by a magnetic field with a frequency of (**a**)—*f*_0_/2 and (**b**)—*f*_0_/3.

**Figure 7 sensors-23-04523-f007:**
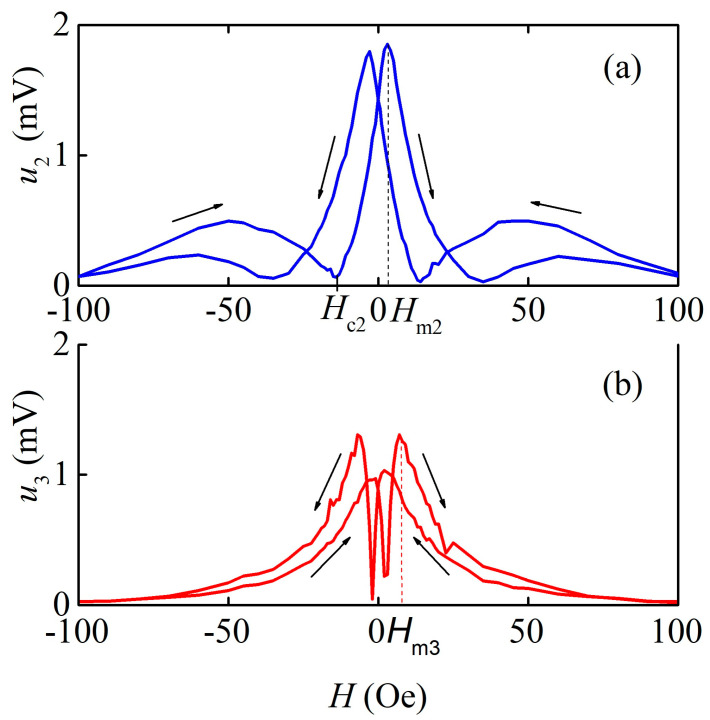
Dependences of the second (**a**) and third (**b**) harmonics amplitudes of the ME voltage *u*_2_ (at *h* = 0.11 Oe) and *u*_3_ (at *h* = 0.16 Oe) on the permanent field *H*. Arrows denote the direction of field change.

**Figure 8 sensors-23-04523-f008:**
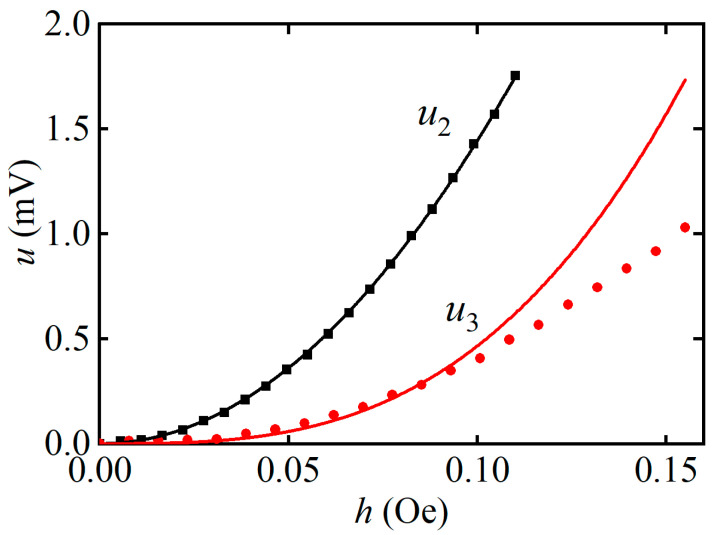
Dependencies of the amplitudes of the second and third harmonic of the ME voltage *u*_2_ and *u*_3_ on the amplitude of the excitation magnetic field *h*. The dots show experiment data. The lines are approximations by power functions.

**Figure 9 sensors-23-04523-f009:**
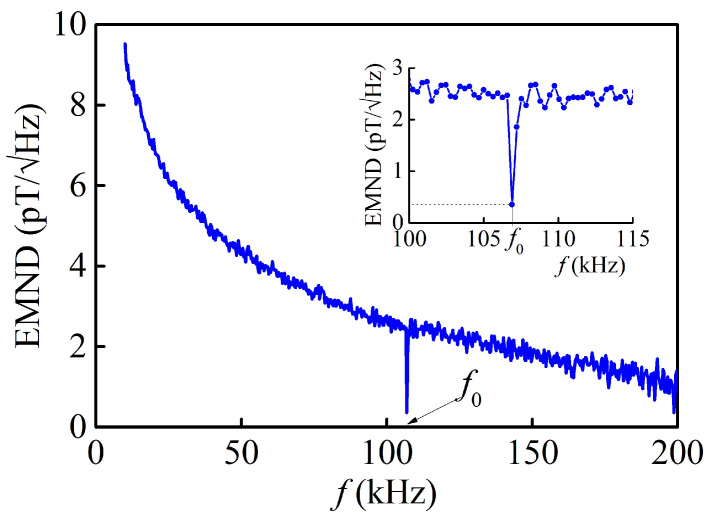
EMND as a function of frequency *f*. The inset shows the resonance region on an enlarged scale.

**Table 1 sensors-23-04523-t001:** Sensitivity of ME structures.

ME Structure	Sensitivity, V/Oe	Reference
AlN/Metglas	1	[23]
AlScN/FeCoSiB	0.13	[24]
LGT/Metglas	1	[25]
PMN-PT/Metglas	9.5	[10]
Quartz/Metglas	0.29	[26]
bidomain LiNbO_3_/Metglas	3.94	[27]
LGT/FeCoSiB film	4.4	This work

## Data Availability

The data that support the findings of this study are available from the corresponding author upon reasonable request.

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
