# Peer review of "Magnetoelectric Effect in Amorphous Ferromagnetic FeCoSiB/Langatate Monolithic Heterostructure for Magnetic Field Sensing"

_sensors, 2023, doi:10.3390/s23094523_

Round 1

Reviewer 1 Report

The authors reported a heterostructure for magnetic field sensing. The ferromagnetic layer FeCoSiB was deposited by magnetron sputtering on the piezoelectric Langatate substrate. The results of linear and nonlinear magnetoelectric effect of heterostructure were characterized. Publication in Sensors could be considered if the authors carefully address the following comments.

1. What are the advantages of magnetic sensor based on magnetoelectric effect? I suggest the authors to give out some examples or references.

2. The authors only compare the sensitivity of their heterostructure with that of other heterostructures. What about the other parameters? To my understanding, there should be other important parameters.

3. I strongly suggest the authors to compare their heterostructure magnetic sensor based on magnetoelectric effect with other kinds of magnetic sensors, for example, magnetic sensors based on Hall effect in semiconductors, and so on. This is good for readers to understand the advantages of magnetoelectric effect based magnetic sensor.

4. What are the influences of thicknesses of piezoelectric and ferromagnetic layers? Why are the ferromagnetic layers deposited on both sides of piezoelectric layer?

5. Which factors determine the resonant frequency of heterostructure? Is it the higher the better?

Minor editing of English language required.

Author Response

Reviewer #1 (Comments and Suggestions for Authors)

The authors reported a heterostructure for magnetic field sensing. The ferromagnetic layer FeCoSiB was deposited by magnetron sputtering on the piezoelectric Langatate substrate. The results of linear and nonlinear magnetoelectric effect of heterostructure were characterized. Publication in Sensors could be considered if the authors carefully address the following comments.

  1. What are the advantages of magnetic sensor based on magnetoelectric effect? I suggest the authors to give out some examples or references.

Answer:

The main advantages of magnetic field sensors based on the magnetoelectric effect are their high sensitivity, reaching a few V/Oe, and their ability to detect small fields with a detection limit of the order of pT/Hz1/2 at room temperature. For comparison, SQUID magnetometers operate at temperatures below room temperature, and currently, the most common Hall sensors have a detection limit of the order of a few nT/Hz1/2 [1-3 (from manuscript)].

  1. The authors only compare the sensitivity of their heterostructure with that of other heterostructures. What about the other parameters? To my understanding, there should be other important parameters.

Answer:

The heterostructures proposed in this paper are promising for creating sensors for alternating magnetic fields with low amplitudes. The most important characteristics of such sensors are sensitivity, minimal detectable fields, and noise level. We compared the sensitivity in the article with that of other heterostructures. The minimal detectable field was provided in the text of the article (line 239); in other works, this value is rarely indicated. We also compared the level of magnetic noise (EMND) with that of noise in other structures (lines 306-312). Temperature characteristics are also important, but their study was not a part of this work.

  1. I strongly suggest the authors to compare their heterostructure magnetic sensor based on magnetoelectric effect with other kinds of magnetic sensors, for example, magnetic sensors based on Hall effect in semiconductors, and so on. This is good for readers to understand the advantages of magnetoelectric effect based magnetic sensor.

Answer:

Let us compare the obtained results with those of commercially available and widely used sensors, such as the Analog Devices AD22151 Hall sensor and the Bartington Mag 646 fluxgate. The parameters of the studied sensors are given in the table. The sensor based on the magnetoelectric effect is capable of detecting magnetic fields with amplitudes several orders of magnitude smaller than those of the Hall sensor and the fluxgate, while having much higher sensitivity compared to the Hall sensor and comparable sensitivity to the fluxgate.

Parameters

Hall sensor Analog Devices AD22151

Fluxgate Bartington Mag 646

This work

Limit of detection, T

10-5

1·10-7

70·10-12

Sensitivity, V/Oe

0.0004

10 (100 mV/μT)

4.6

Noise level, pT/Hz1/2

-

25

0.36

  1. What are the influences of thicknesses of piezoelectric and ferromagnetic layers? Why are the ferromagnetic layers deposited on both sides of piezoelectric layer?

Answer:

Both experimentally and theoretically, the influence of layer thicknesses on the magnitude of the ME effect has been studied in many works, for example: [doi: 10.1109/TUFFC.2003.1244741, 10.1007/s11433-011-4268-2, 10.1063/1.3222914, 10.1088/1361-665X/abf6c0, 10.21883/FTT.2020.08.49600.049, 10.1134/1.1946866]. It has been shown that the maximum ME effect is achieved at a certain ratio of thicknesses and Young's moduli of the materials [doi: 10.21883/FTT.2020.08.49600.049, 10.1134/1.1946866]. An increase in the thickness of the ferromagnetic layer also leads to an increase in the influence of the demagnetizing factor, which in turn leads to an increase in the field where the highest ME coefficients are observed. In this work, ferromagnetic layers were also used as electrodes. A feature of such a symmetrical structure is that it effectively suppresses bending vibrations [doi: 10.1103/PhysRevB.86.214405].

  1. Which factors determine the resonant frequency of heterostructure? Is it the higher the better?

Answer:

The resonance frequency of longitudinal oscillations is determined by the formula [27 (from manuscript)]:

,

where L is the length of the sample, ,  are the effective Young's modulus and density of the structure, respectively. Thus, the resonance frequency depends on the length of the heterostructure (the longer it is, the lower the resonant frequency), the effective Young's modulus, and the density of the structure. Both parameters are defined as:

,

.

Here, the indices p and m denote, respectively, the piezoelectric and magnetostrictive layers, and t is the thickness of the corresponding layers. Thus, the resonance frequency is also dependent on the mechanical parameters of the layer materials used and their dimensions.

Since the sensors considered in the ME work operate in the resonance mode, their maximum sensitivity is achieved in a narrow frequency band near the resonance frequency. Depending on the task, by varying the parameters of the heterostructures, it is possible to shift the resonant frequency in a wide range.

Reviewer 2 Report

This manuscript has reported the possibilities of creating magnetic field sensors using the direct ME effect in a monolithic heterostructure of amorphous ferromagnetic material/langatate. However, there are still some problems that authors needed to improve. 

1. The introduction part seems not enough. The authors needed to do more literature reviews to expand the background.

2. For ME, there are several ways in which a magnetoelectric effect can arise in a material. The authors need to explain it in detail. Like Single-ion anisotropy, or Symmetric Exchange striction?

Moderate editing of English language

Author Response

Reviewer 2.

This manuscript has reported the possibilities of creating magnetic field sensors using the direct ME effect in a monolithic heterostructure of amorphous ferromagnetic material/langatate. However, there are still some problems that authors needed to improve. 

  1. The introduction part seems not enough. The authors needed to do more literature reviews to expand the background.

Answer:

We added to the introduction the following discussion:

"In recent work [doi: 10.1063/5.0124352], the behavior of organic and inorganic glues for high-temperature operations of an ME sample based on a 0.364BiScO3–0.636PbTiO3 piezoelectric ceramic and Terfenol-D alloy was investigated. It was shown that the organic glue-based ME composite sharply decreases the output signal with increasing temperature, while the inorganic glue-based ME composite gradually decreases the ME effect (by 20% less at 200°C than at room temperature). The use of sputtering technique for the ME composite structure eliminates losses related to glue or other connecting layers."

  1. For ME, there are several ways in which a magnetoelectric effect can arise in a material. The authors need to explain it in detail. Like Single-ion anisotropy, or Symmetric Exchange striction?

Answer:

In article (line 37-39) we wrote:

“The ME effect manifests itself in the generation of an electric voltage by the structure under the action of a magnetic field resulting from the deformation of the FM layer and, as a consequence, polarization of the mechanically linked PE layer [4].”

Here works separately well-known effects magnetostriction in ferromagnetic material and piezoelectric effect, their combination caused a magnetoelectric effect.